# Knowledge, attitude and practice of cervical cancer screening among women infected with HIV in Africa: Systematic review and meta-analysis

Agajie Likie Bogale[1]*, Tilahun Teklehaymanot[2], Jemal Haidar Ali[3], Getnet Mitike Kassie[4]

1 Ethiopian Public Health Institute, Addis Ababa University, Addis Ababa, Ethiopia, 2 Aklilu Lemma Institute of Pathobiology, Addis Ababa University, Program of Tropical and Infectious Diseases, Addis Ababa, Ethiopia, 3 Addis Ababa University, College of Health Sciences, School of Public Health, Addis Ababa, Ethiopia, 4 International Institute for Primary Health Care-Ethiopia, Addis Ababa, Ethiopia

* agalb.2000@yahoo.com

## Abstract

**Data Availability Statement:** All relevant data are within the manuscript and its Supporting Information files.

### Background

To establish successful strategies and increasing the utilization of preventive services, there is a need to explore the extent to which the general female population is aware and use the service for cervical cancer-screening among women infected with HIV in Africa. Available evidences in this regard are controversial and non-conclusive on this potential issue and therefore, we estimated the pooled effect of the proportion of knowledge, attitude and practice of HIV infected African women towards cervical cancer screening to generate evidence for improved prevention strategies.

### Methods

We applied a systematic review and meta-analysis of studies conducted in Africa and reported the proportion of knowledge, attitude and practice towards cervical cancer screening. We searched electronic databases: PubMed/Medline, SCOPUS, ScienceDirect, Web of science, Cumulative Index of Nursing and allied Health Sciences (CINAHL) and Google scholar databases to retrieve papers published in English language till August 2020. We used random-effects model to estimate the pooled effect, and funnel plot to assess publication bias. The registration number of this review study protocol is CRD42020210879.

### Results

In this review, we included eight published papers comprising 2,186 participants. The estimated pooled proportion of knowledge of the participants was 43.0% (95%CI:23.0–64.0) while the pooled estimates of attitudes and practices were 38.0% (95%CI: 1.0–77.0) and 41.0% (95%CI: 4.0–77.0), respectively. The proportion of the outcome variables were extremely heterogeneous across the studies with $I^2 > 98\%$).

**Funding:** The authors received no specific funding for this work.

**Competing interests:** The authors have declared that no competing interests exist.

**Abbreviations:** CI, Confidence interval; df, degree of freedom; HIV, Human Immunodeficiency virus; $I^2$, Heterogeneity; KAP, Knowledge, attitude and practice; MeSH, Medical subject heading; tiab, Title and abstract.

## Conclusion

The pooled estimates of knowledge, attitude and practice were lower than other middle income countries calls for further activities to enhance the uptake of the services and establish successful strategies.

## Introduction

Cancer of the cervix uteri is the 3rd most common cancer among women worldwide, with an estimated 569,847 new cases and 311,365 deaths with a greater number of cases (119,284) and deaths (81,687) in Africa, according to the(GLOBOCAN 2018, an online database providing estimates of incidence and mortality) [1]. This death report is even higher than worldwide report in 2012 indicating that 266,000 women died of cervical cancer–equivalent of one woman dying every 2 minutes with about 90% of these deaths occurring in low- and middle-income countries [2].

Cancer of the cervix is the second most commonly diagnosed cancer after breast cancer and the third leading cause of cancer death after breast and lung cancers in developing countries [3]. It also ranks second next to breast cancer in Ethiopia [4].

One of the strategies to minimize the burden of the disease is to establish successful strategies and increasing the utilization of preventive measures ranging from community education, social mobilization, vaccination, screening, and treatment to palliative care [5].

Most importantly, increasing the knowledge, attitude and practice (KAP) of cervical cancer screening and prevention among females is a part of a comprehensive approach to cervical cancer prevention and control strategy. This might play a pivotal role in the controlling strategy on the issue. Health workers considered to take a lead in this regard are found to have less knowledge about cervical cancer as a disease and relatively fair knowledge on Pap smear testing [6]. Among women who had been attending a tertiary hospital, the majority had a positive attitude while about a third had good knowledge and very few (2.7%) had good practice [7]. When looked at the patients suffered gynecological cancer, more than half of them knew that their disease was malignant [8]. In addition, different evidence about knowledge, attitude and practice were generated at different corners of Asian countries [9–12] that seek for pooling of the findings for decision making.

In Africa, the findings indicate that the cervical cancer screening approach is in its infancy stage. For instance, in rural Uganda, only 4.8% of women had ever been screened for cervical cancer [13] and around ten percent in Burkina-Faso [14]. In Ethiopia, including University female students, their KAP is fair towards cervical cancer and scored less than fifty percent [15–18].

On the basis of the comprehensive literature search made, variability on the KAP score prevails in various African countries and assumed to have high prevalence of the problem and with unavailability of information among the female population living with HIV who are the most vulnerable population.

Therefore, this review aims to estimate the pooled effect of the proportion of KAP of HIV infected African women towards cervical cancer screening to generate evidence for improved prevention strategies.

## Materials and methods

### Search strategy and screening of papers

We conducted a systematic review and meta-analysis of published articles to estimate the pooled effect or the proportion of knowledge, attitude and practice towards cervical cancer

screening in Africa. We systematically searched the papers published in the following electronic databases; PubMed/Medline, SCOPUS, ScienceDirect, Web of science, Cumulative Index of Nursing and allied Health Sciences (CINAHL) and Google scholar. The review was conducted in accordance with Preferred Reporting Items for Systematic Reviews and Meta-Analyses (PRISMA) standard [19] as displayed in S1 Table. We used a search strategy by combining the following key terms: knowledge, attitude, practice, cervical cancer, uterine cervical neoplasms, cervical cancer screening, human immunodeficiency virus or HIV, and Africa. We used Truncation(*) to manage spelling variation during search: infect* or positive, wom*n or female* or girl*. We used both free text and Medical subject heading [MeSH]terms during electronic database search.

PubMed database search strategy was: (((((((knowledge) AND ((((cervical cancer) OR (Uterine Cervical Neoplasms)) OR (cervical cancer screening[tiab])) OR (cervical cancer screening[MeSH Terms]))) AND ((Attitude) AND ((((cervical cancer) OR (Uterine Cervical Neoplasms)) OR (cervical cancer screening[tiab])) OR (cervical cancer screening[MeSH Terms])))) AND ((practice) AND ((((cervical cancer) OR (Uterine Cervical Neoplasms)) OR (cervical cancer screening[tiab])) OR (cervical cancer screening[MeSH Terms])))) AND ((human immunodeficiency virus) OR (HIV))) AND ((infect*) OR (positive))) AND ((women) OR (female*))) AND (Africa) AND ((y_10[Filter]) AND (female[Filter]) AND (english[Filter]))

The search was repeated to identify the consistency of search terms and result. Two authors (AL and JH) independently reviewed the titles, abstracts and full articles of retrieved studies.

## Study inclusion and exclusion criteria

In this review, we included a cross sectional studies conducted in Africa that reported Knowledge, attitude and practice towards cervical cancer screening. The inclusion was restricted to papers published from 2010 to August 2020 in the English language within a ten year period though data were available from 2014 to 2019. We excluded those studies that did not clearly state the outcome measures, study population different from HIV infected women or females, duplication citations, and review articles [**Fig** 1].

## Study quality assessment

We assessed the quality of included studies by using the 14 items Quality Assessment Tool for Observational Cohort and Cross-Sectional Studies—NHLBI, NIH [20]. This assessment tool mainly focused on research question, study population, eligibility criteria (inclusion and exclusion criteria of study participants), sample size justification, exposure measures and assessment, sufficient time frame to see an effect, outcome measures and blinding of outcome assessors, follow up rate, and statistical analysis. The quality assessment was rated as good, fair and poor based on quality assessment tool criteria. The maximum score indicating high quality was 14 and the lowest possible score was zero. The rating values of the included studies in terms of their quality were based on their design. Cross-sectional types do not consider the items which fit for cohort and taken as not-applicable (NA) and thus, the rating values were not taken from the possible maximum score (i.e. 14). In this review, all scores are written in percentage beside the results individual components of the quality assessment [S2 Table].

## Data extraction

We extracted data from eligible abstract and/or full text of the articles by considering the outcome variables and the characteristics of participants such as age range, mean or median age, sex, HIV sero-status. In addition, we extracted the study characteristics such as first author,

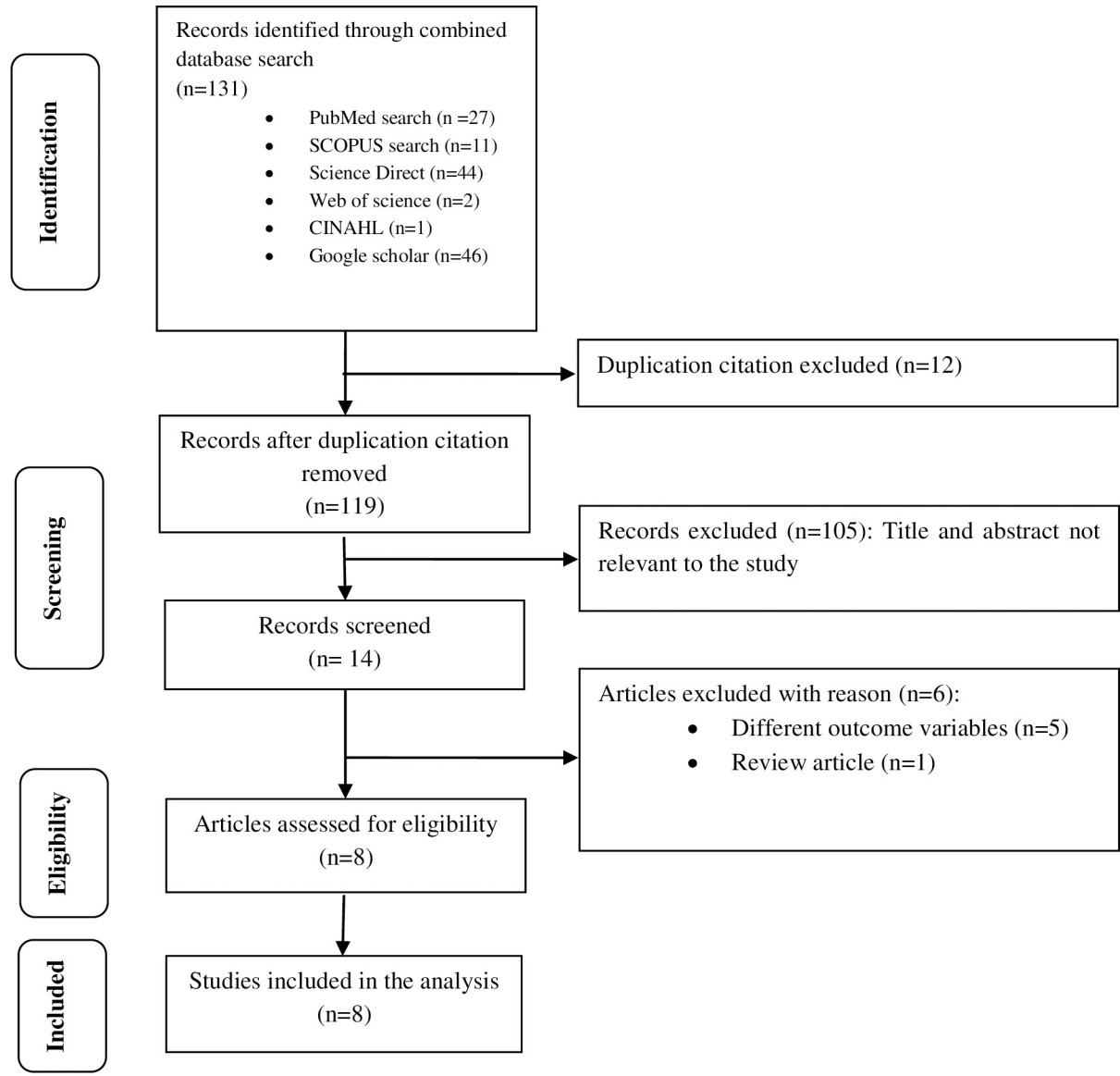

**Fig 1. Flow diagram of studies reviewed, screened and included.**

year of publication, study setting, study location or country, study design, sample size, knowledge score, attitude and practice [**Table 1**].

## Statistical analysis

We estimated the pooled proportion of knowledge, attitude and practice of HIV positive women on cervical cancer screening with its 95% Confidence Interval (CI) using random effects meta-analysis model assuming the true effect size varies between studies [21]. The proportion of knowledge, attitude and practice reported in each included study is multiplied by its sample size to express the score in number, and data presented in forest plot.

We assessed heterogeneity in the proportions of different studies using heterogeneity Chi-square ($x^2$) based Q test with significant level of p-value $< 0.1$ and $I^2$. The $I^2$ value 25% indicates low heterogeneity while 50% moderate and 75% high [22]. We assessed the potential

**Table 1. Characteristics of included studies to estimate the pooled effect of knowledge, attitude and practice of HIV-infected women towards cervical cancer in Africa.**

| First Author | Year | Study setting | Study location | Study design | Sample size | Knowledge | Attitude | Practice | Age range/mean age in years |
|---|---|---|---|---|---|---|---|---|---|
| Solomon et al [24] | 2019 | Health facility | Ethiopia | Cross-sectional | 475 | | | 119 | 36 |
| Shiferaw et al [25] | 2018 | Health facility | Ethiopia | Cross-sectional | 581 | 136 | | | 35 |
| Mitchell et al [26] | 2017 | Health facility | Uganda | Cross-sectional | 87 | | 1 | | 30–69 |
| Stuart et al [27] | 2019 | Health facility | Ghana | Qual-Quantof parent cohort | 60 | 48 | | | > = 18 |
| Adibe & Aluh [28] | 2017 | Health facility | Nigeria | Cross-sectional | 447 | 45 | 194 | | > = 9 |
| Belglaiaa et al [29] | 2018 | Health facility | Morocco | Cross-sectional | 115 | 24 | | 15 | 34.9 |
| Rosser et al [30] | 2015 | Health facility | Kenya | Cross-sectional | 106 | 69 | 74 | 89 | 34.9 |
| Maree & Moitse [31] | 2014 | Health facility | South Africa | Cross-sectional | 315 | 198 | | | 38.9 |

The proportions of knowledge, attitude and practice is written in number to make the data which fit for meta-analysis using metaprop.

publication bias using funnel plot. If the 95% of the point estimates of the included studies lie within the funnel plot defined by straight lines, then that indicates the absence of heterogeneity [23]. We used moment based meta-regression to assess the potential source of heterogeneity. Data analysis was conducted using STATA version 14.

**Ethical approval and consent to participate.** Since the review made was based on previously published articles, there was no need for ethical clearance. Nevertheless, the protocol of the study was pre-registered on PROSPERO (International prospective register of systematic reviews) University of York, Centre for Reviews and Dissemination with registration number CRD42020210879.

**Operational definition of KAP in this protocol.** *Knowledge (K)*. Refers to the awareness of HIV positive women towards cervical cancer screening in Africa. Different pocket studies are filtered and eligible articles are included in the analysis to estimate the pooled knowledge.

*Attitude (A)*. Refers to the way of thinking or feeling of HIV positive women on cervical cancer screening.

*Practice (P)*. Refers to the habit of women to be screened for cervical cancer.

# Results

## Study characteristics

We included eight studies [Fig 1], from Ethiopia [24, 25], Uganda [26], Ghana [27], Nigeria [28], Morocco [29], Kenya [30], and South Africa [31] which are health facility based [Table 1]. Almost all the included studies were cross-sectional types published from 2014 to 2019 though the extraction of data was done for the past ten years till August 2020. The maximum sample size reported was 581 [25] while the minimum was 60 [27]. The age of respondents ranged from 9 to 69 years [Table 1].

## Pooled estimates of knowledge of HIV positive women towards cervical cancer screening in Africa

We pooled data from 2,186 HIV positive women to estimate the pooled proportion of knowledge on cervical cancer screening using meta-analysis. The overall pooled proportion of knowledge was 43.0% [Fig 2] with high heterogeneity across the studies(chi$^2$ = 493.23 (d.f. = 5), p = 0.001, and I$^2$ = 98.99%).

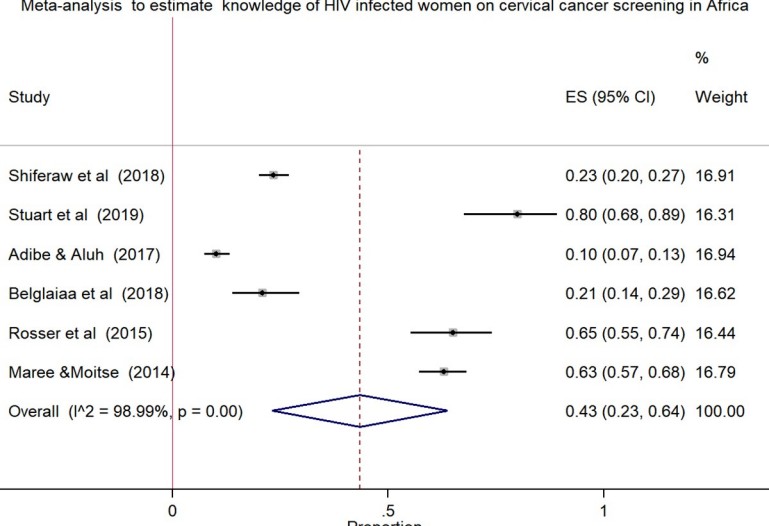

**Fig 2. Forest plot to estimates the proportion of knowledge among HIV infected women towards cervical cancer screening in Africa with 95% CI (the estimate weighted based on random effects model): ES-Effect size equivalent to the proportion, CI-Confidence interval.** In the plot, the diamond shows the pooled result and the boxes show the effect estimates from the single studies. The purple dotted vertical line indicates pooled estimate. The purple solid vertical line indicates the reference line at zero indicating no effect. The horizontal line through the boxes illustrate the length of the confidence interval and the boxes show the effect estimates from the single studies.

## Pooled estimates of the attitude and practice of HIV positive women towards cervical cancer screening in Africa

Meta-analysis using few studies included is not recommended due to less precision. In this review, only three studies included in the analysis to estimate the pooled effect, which was the attitude and practice of HIV positive women towards cervical cancer screening. The pooled estimates of attitude was 38.0% [Fig 3] with high heterogeneity across the studies ($chi^2 = 437.57$ (d.f. = 2), p = 0.001, $I^2$ (variation in ES attributable to heterogeneity) = 99.54%) while the pooled practice estimate was 41.0% [Fig 4] with high level of heterogeneity ($chi^2 = 260.80$ (d.f. = 2), p = 0.001, $I^2$ (variation in ES attributable to heterogeneity) = 99.23%).

## Meta-regression analysis

We assessed the effect of sample size and year of the study on heterogeneity between the studies using meta-regression model. However, there was no significant prediction of heterogeneity between the effect size and the assessed variables (i.e., both sample size and year of the study) [Table 2]. Meaning, in the adjusted model, both sample size and year of the study didn't indicate heterogeneity in the effect size which is equivalent to the pooled proportion (P > 0.05). When we interpret the finding using β-coefficient, one unit increase in the publication year will decrease the outcome variable by the coefficient of 20.14 points and an increase in the sample size will depict a slight increase (0.14 points) in the outcome [Table 2].

## Publication bias

The funnel plot (widely used to examine bias in the result of a meta-analysis) for pooled estimates of knowledge, attitude and practice towards cervical cancer screening indicated that there was publication bias. The included studies are scattered out of pseudo 95% confidence limit and the observed bias might be due to small study effect [Fig 5A–5C]. Fig 5A, indicates

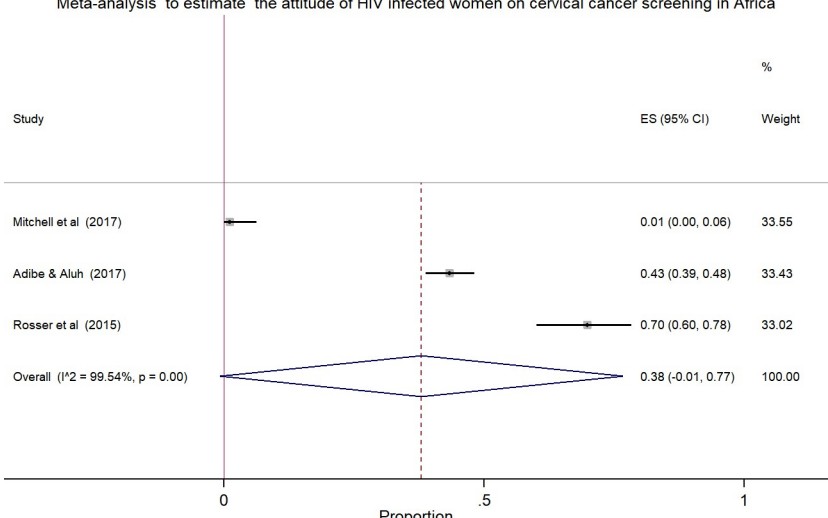

**Fig 3. Forest plot to estimates the proportion of attitude among HIV infected women towards cervical cancer screening in Africa with 95% CI (the estimate weighted based on random effects model): ES-Effect size equivalent to the proportion, CI-Confidence interval.** In the plot, the diamond shows the pooled result and the boxes show the effect estimates from the single studies. The purple dotted vertical line indicates pooled estimate. The purple solid vertical line indicates the reference line at zero indicating no effect. The horizontal line through the boxes illustrate the length of the confidence interval and the boxes show the effect estimates from the single studies.

funnel plot of the 6 estimates of knowledge (k) towards cervical cancer screening available for meta-analysis (SE-Standard error, ES-Effect size: proportion), (**b**) Funnel plot of the 3 estimates of attitude (A) towards cervical cancer screening available for meta-analysis (SE-Standard error, ES-Effect size: proportion), (**c**) Funnel plot of the 3 estimates of practice towards

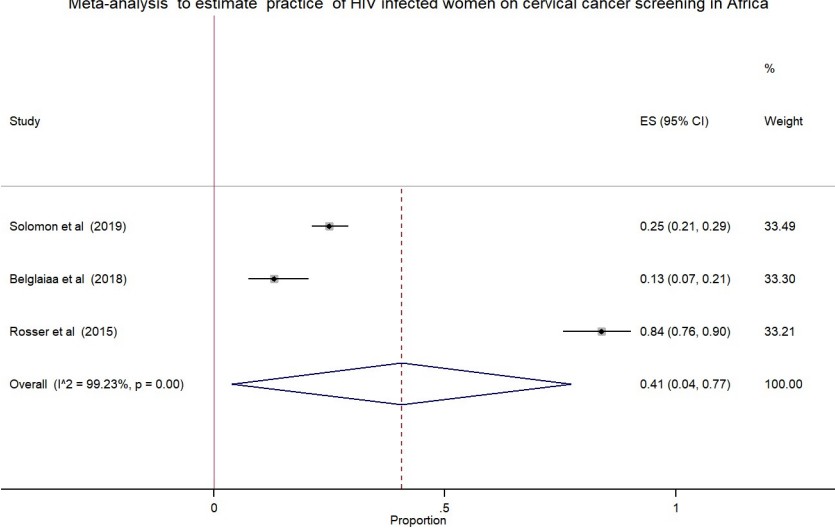

**Fig 4. Forest plot to estimates the proportion of practice among HIV infected women towards cervical cancer screening in Africa with 95% CI (the estimate weighted based on random effects model): ES-Effect size equivalent to the proportion, CI-Confidence interval.** In the plot, the diamond shows the pooled result and the boxes show the effect estimates from the single studies. The purple dotted vertical line indicates pooled estimate. The purple solid vertical line indicates the reference line at zero indicating no effect. The horizontal line through the boxes illustrate the length of the confidence interval and the boxes show the effect estimates from the single studies.

**Table 2. Meta-regression analysis for year of study and sample size as a reason of heterogeneity on the knowledge estimates of women infected by HIV in Africa.**

| Variable | Adjusted model | | |
|---|---|---|---|
| | ß(95% CI) | SE | P-value |
| Sample size | 0.14(-0.24–0.52) | 0.12 | 0.32 |
| Publication year | -20.14(-61.7–21.4) | 13.1 | 0.22 |

SE-Standard error, ß-regression coefficient, 95% CI Confidence interval

cervical cancer screening available for meta-analysis (SE-Standard error, ES-Effect size: proportion).

In this plot, the blue broken line indicates Pseudo 95% CI, the solid red line indicates pooled estimate of the proportion of knowledge, attitude and practice, and the scattered circle dots indicates included studies in the meta-analysis. The scale on the X-axis indicates Effect size estimate or proportion and the Y-axis indicates the precision estimate using standard Error.

## Discussion

In this review, the pooled estimate of knowledge, attitude and practice of HIV infected women towards cervical cancer screening in Africa was 43.0%, 38.0% and 41.0%, respectively. The highest heterogeneity and publication bias were observed in meta-analysis using forest plot and funnel plot, respectively. The Meta-regression model was applied to identify the reason for heterogeneity using sample size and publication years. However, the variation did not show a significant association on the effect size equivalent to the proportion or the outcome variables.

The knowledge estimate of our finding was concordant with previously reported review findings in Ethiopia among women of reproductive age group (40.37%) [32] while the attitude and practice findings varied and were 58.87% and 14.02%, respectively [32]. Such variations are likely due to the fact that the study population and settings studied were different from our current review focused on HIV positive women in Africa.

Similarly, Kasraeian et al also in their review made in low and middle-income countries indicated that HIV positive women had less knowledge about cervical cancer and were less likely to undergo screening [33].

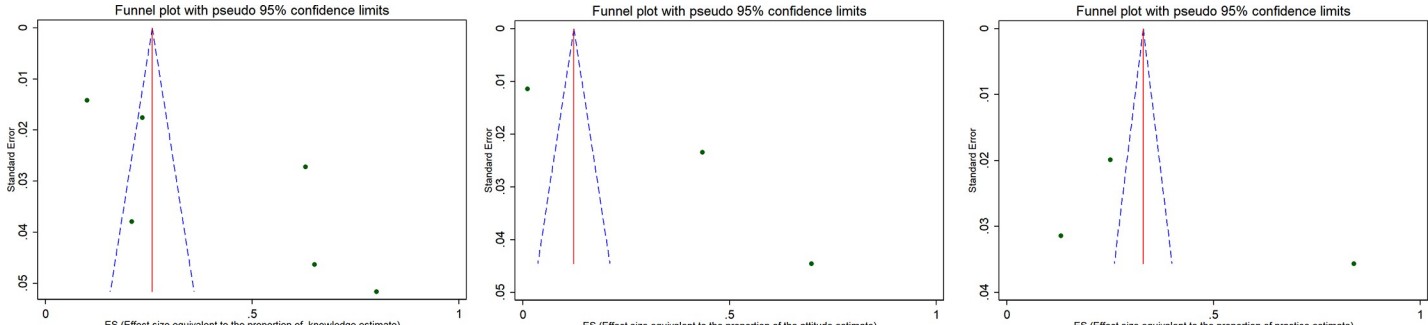

**Fig 5. Publication bias assessment: a) Funnel plot of the 6 estimates of knowledge (k) towards cervical cancer screening available for meta-analysis (SE-Standard error, ES-Effect size: proportion), b) Funnel plot of the 3 estimates of attitude (A) towards cervical cancer screening available for meta-analysis (SE-Standard error, ES-Effect size: proportion), c) Funnel plot of the 3 estimates of practice towards cervical cancer screening available for meta-analysis (SE-Standard error, ES-Effect size: proportion).** In this plot, the blue broken line indicates Pseudo 95% CI, the solid red line indicates pooled estimate of the proportion of knowledge, attitude and practice, and the scattered circle dots indicates included studies in the meta-analysis. The scale on the X-axis indicates Effect size estimate or proportion and the Y-axis indicates the precision estimate using standard Error.

The original research articles conducted in different Asian countries, including Pakistan illustrated that participants of the study had inadequate knowledge, attitude and practice towards cervical cancer [34]. In the same breath, Iraqi participants only 30.3% of employed and 40.0% of the students of the female population had positive attitude towards cervical cancer screening [35]. Similarly, a study conducted in India indicated that 30.2% of respondents had good knowledge and almost one fourth (25.9%) had a favorable attitude towards cervical cancer [9] with an inspiring result of 58.9% documented for female health care providers' knowledge of cervical cancer screening [10]. Another study adds up, only 36.48% of the participants have good knowledge and of these, 83.78% of them had a positive attitude, though the vast majority (97.29%) had no practice [7] revealing a big gap between attitude and practice. The study conducted in the Eastern China also reported slightly over half (51.9%) of rural women to have high knowledge of whom, 96.0%of them expressed positive attitude and 63.7% were screened for cervical cancer [11]. Another study from Nepal also reported 42.9% of women to have knowledge and more than 85.0% to have had a positive attitude towards cervical screening [12]. Whereas the Iranian study showed more than half (58.0%) of patients with cancer knew that their diseases was malignant [8].

When looked at the African continent, very diverse findings were also reported. According to the Tanzania study, only 10.4% of women were knowledgeable about cervical cancer and 7.9% of these were screened [36] which was very low. Similarly, in Nigeria18.1% of them had good knowledge with 67.8% of them to have a positive attitude to cervical cancer screening [37]. In Uganda and Burkina-Faso, the two studies reported that only 4.8% and 11.1% of women had been screened for cervical cancer, respectively [13, 14]. The various studies done in Ethiopia also showed different results [15–18]. The study done among University female students in Debre-Berhan, North Ethiopia reported 35.6%, while the study in Wolaita, southern Ethiopia among women of reproductive age reported 43.1% to have good knowledge towards cervical cancer showing better results [15, 16]. Of the 43.1% of the women, studied in Wolaita, 45.5% of them had a positive attitude with 22.9% of them to undergo for screening [16]. On the other hand, the study done in Addis Ababa reported 27.7% of women to have adequate knowledge of cervical cancer and 25.0% of these to undergo for cervical cancer screening [17]. The lowest score for knowledge in Ethiopia was documented for Gonder, northern Ethiopia and it was 19.87% underscoring the need for more advocacy work [18].

Such variable findings of the studies in different countries encouraged us to estimate the pooled effect equivalent to the proportion of the finding which is very crucial for program direction on cervical cancer screening and prevention measure aspects.

## Strength and limitations of the study

The strength of this review is that we attempted to include the most vulnerable population, women living with HIV in Africa, and has captured the recent publications within ten years till August 2020 and used more than five biomedical databases. The limitation however, is the inclusion of those papers published and reported only in English language which might have missed other important works done in this regard. The precision of the pooling effect might have been also affected by the fact that only very few studies which reported attitude and practice were included and this might consequently affect the assertion.

## Conclusion

The pooled estimates of knowledge, attitude and practice of the current review finding was below half in different African countries. The enhancement of knowledge, attitude and practice of women will augment the comprehensive approach to cervical cancer prevention and

control strategy. The current work has shed light-on how much the findings of the studies conducted in different countries on the cervical cancer and its screening were very diverse and difficult for decision making. Thus, it is essential to have the pooled estimates of different findings for decision making. Other than this, the pooled estimates are very crucial for further strengthening the strategies for prevention measure and control of cervical cancer mainly on vulnerable population like women infected by HIV.

## Supporting information

**S1 Appendix. PROSPERO International prospective register of systematic reviews.**
(PDF)

**S1 Table. PRISMA assessment checklist.**
(DOC)

**S2 Table. The results of the individual components of the quality assessment.**
(DOCX)

## Acknowledgments

The authors acknowledge the contribution of Aklilu Lemma Institute of Pathobiology, Addis Ababa University, Program of Tropical and Infectious Diseases and Ethiopian Public Health Institute (EPHI) for the opportunity to access an internet. The unreserved support rendered by Minilik Demesie from EPHI during the reviewing process of the study protocol was highly appreciable.

## Author Contributions

**Conceptualization:** Agajie Likie Bogale.

**Data curation:** Agajie Likie Bogale, Jemal Haidar Ali.

**Formal analysis:** Agajie Likie Bogale.

**Investigation:** Agajie Likie Bogale, Jemal Haidar Ali.

**Methodology:** Agajie Likie Bogale, Tilahun Teklehaymanot, Jemal Haidar Ali, Getnet Mitike Kassie.

**Project administration:** Agajie Likie Bogale.

**Writing – original draft:** Agajie Likie Bogale.

**Writing – review & editing:** Tilahun Teklehaymanot, Jemal Haidar Ali, Getnet Mitike Kassie.

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
