## [Decision Letter · Decision Letter 0]

4 Mar 2021

PONE-D-21-03081

Knowledge, attitude and practice of cervical cancer screening among women infected with HIV in Africa: Systematic review and meta-analysis

PLOS ONE

Dear Dr. Bogale

Thank you for submitting your manuscript to PLOS ONE. After careful consideration, we feel that it has merit but does not fully meet PLOS ONE’s publication criteria as it currently stands. Therefore, we invite you to submit a revised version of the manuscript that addresses the points raised during the review process.

Please submit your revised manuscript by 3/27/2021 If you will need more time than this to complete your revisions, please reply to this message or contact the journal office at plosone@plos.org. Please include the following items when submitting your revised manuscript:

We look forward to receiving your revised manuscript.

Kind regards,

Saeed Ahmed, MD

Academic Editor

PLOS ONE

2. Please confirm that you have included all items recommended in the PRISMA checklist including:

-    the full electronic search strategy used to identify studies with all search terms and limits for at least one database.

-    an explanation for why the search inclusion dates were only for the past 10 years

-    a Supplemental file of the results of the individual components of the quality assessment, not just the overall score, for each study included.

-    See https://journals.plos.org/plosmedicine/article?id=10.1371/journal.pmed.1000100#pmed-1000100-t003 for guidance on reporting.

Thank you.

3. Please include your tables as part of your main manuscript and remove the individual files. Please note that supplementary tables (should remain/ be uploaded) as separate "supporting information" files.

4.Thank you for stating the following financial disclosure:

Reviewers' comments:

Reviewer's Responses to Questions

**Comments to the Author**

1. Is the manuscript technically sound, and do the data support the conclusions?

Reviewer #1: Yes

Reviewer #2: Yes

Reviewer #3: Yes

Reviewer #4: Partly

Reviewer #5: Partly

2. Has the statistical analysis been performed appropriately and rigorously? 

Reviewer #1: Yes

Reviewer #2: Yes

Reviewer #3: I Don't Know

Reviewer #4: Yes

Reviewer #5: Yes

3. Have the authors made all data underlying the findings in their manuscript fully available?

Reviewer #1: Yes

Reviewer #2: Yes

Reviewer #3: Yes

Reviewer #4: Yes

Reviewer #5: Yes

4. Is the manuscript presented in an intelligible fashion and written in standard English?

Reviewer #1: No

Reviewer #2: Yes

Reviewer #3: Yes

Reviewer #4: Yes

Reviewer #5: Yes

5. Review Comments to the Author

Reviewer #1: Dear Authors,

This is a systematic review and meta-analysis regarding the knowledge, attitude and practice towards cervical cancer screening in Africa. The study included eight published papers and found that the estimated pooled proportion of knowledge, attitudes and practices among the participants were 43%, 38% and 41% respectively.

Major strength of the study was the large number of participants from the pooled review and statistical analysis. Weaknesses include only eight published papers from a large time frame which is not well described.

There are quite a number of grammatical errors.

I have added some of the grammatical errors and sentence formation that needs to be revised. Some areas are not clear as to what the authors are trying to convey.

Line 31 To establish successful strategies and increasing the utilization of preventive services

Line 38 Methods- Would help to have definite time frame- says till august 2020 but does not say from when

Line 50 Conclusion: The pooled estimates of knowledge, attitude and practice were lower than other developed countries calls for further activities to enhance the uptake of the services and establish successful strategies.

Line 79 When looked at the patients suffered gynecological cancer, more than half of them knew that their disease was malignant

Line 80 In addition, different evidence about knowledge, attitude and practice were generated at different corners of Asian countries [9] [10] [11] [12] that seek for pooling of the findings for decision making.

Line 88 On the basis of the comprehensive literature search made, variability on the KAP score prevails in various African countries assumed have high prevalence of the problem and moreover unavailability of information among the female population living with HIV who are the most vulnerable population.

Line 119 In this review, we included a cross sectional studies conducted in Africa

Line 120 The inclusion was restricted to papers published in the English language within a ten year period till last August 2020

Page 6- identification tab states n=131 and gives the number of articles in each database but they do not add up to 131. They only add up to 85.

Line 168-170 We extracted data from eligible abstract and/or full text of the articles by considering the outcome variables and the characteristics of participants such as age range, mean or median age, sex, HIV sero-status.

Line 193 Almost all included studies were cross-sectional types published from 2014 to 2019 though the extraction of data was done for the past ten years till August 2020.

Reviewer #2: Study is a Systematic review and meta-analysis, with no conflict of interests from authors and no conflict from a funding perspective. Published articles were from well accepted and vetted databases in accordance with PRISMA. Authors included cross sectional studies conducted in Africa that reported Knowledge, attitude and practice towards cervical cancer screening in HIV. Study has included 131 articles based on initial screening and after applying Quality Assessment Tool for Observational Cohort and Cross-Sectional Studies, total 8 articles were analyzed.

Study, very importantly is addressing the key aspect to devise a well though out prevention strategy: the knowledge, attitude and practice (KAP) of a particular subgroup. The study goal is especially important as the group is the female population living with HIV who are the most vulnerable population. Documentation of KAP among HIV infected African women towards cervical cancer screening will have positive effects towards the goal to decrease HIV prevalence, viral suppression HIV as well as the comorbidity of Cervical Cancer. I think it was an appropriate to restrict the studies done within the last 10 years as we are trying to assess the current KAP.

Minor revision Recommendations.

1. 8 studies evaluated where all healthcare facility based. It will be appropriate to know what kind of facility. For example, if all the patients were from a tertiary hospital vs rural health center, I will worry about disparity in healthcare access creating a bias on KAP. Can the authors clarify that with an additional chart?

2. Most patients were from Ethiopia and the difference in health system from different countries can affect the generalizability of the result to whole of the African Continent. Can that be clarified? If not mentioned as a concern in the discussion.

3. Study team could have included CD4 as well as AIDS terms to their search strategy key terms broadening some data collection keeping immune suppression in mind. Cervical cancer incidence correlate with a lower CD4 count. ( JAMA. 2005;293:1471–1476.)

Reviewer #3: Well written paper with nice introduction, aim, and clear methodology, few points that i feel should be addressed please:

- The author need to define in the methodology what they meant by knowledge, attitude and practice (KAP)

- Page 2, line 42, no need to say last August 2020, August 2020 is enough

- Page 3, line 70 add measures after preventive

- Page 5, 2nd paragraph starting with PubMed need to be revised as it include a lot of unnecessary punctuation marks

- Page 5, line 121, again no need to say last August 2020, August 2020 is enough

- Page 13, line 307, again no need to say last August 2020, August 2020 is enough

- The conclusion need to be more precise.

Reviewer #4: - the manuscript needs some minor grammatical/language corrections.

- it is hard to assess the numbers- how do you decide whether knowledge/attitude/practice is appropriate or not? what are the specific parameters you used to decide to decide whether KAP is prevalent or not.

- also as expected there is lot of heterogeneity among the studies.

- in the abstract, the authors have mentioned that KAP is less compared to developed nations, although in the discussion section most comparisons are made to low/middle income and developing nations. Can authors cite and compare their finding with some studies from North American/European/Developed nations

Reviewer #5: This manuscript pooled results about the percentage of knowledge, attitude and practice of cervical cancer screening among women infected with HIV in Africa using meta-analysis. I have below comments.

The overall statistical analysis follows the routine methods of meta-analysis. There are big heterogeneities among studies. It is not necessary to calculate pooled estimates for those results. The heterogeneity may exist because different level of health care, service or education between locations. There might be other factors too. It will be informative to investigate the reason of such existing heterogeneity in knowledge, attitude and practice of cervical cancer screening among women infected with HIV in Africa.

6. PLOS authors have the option to publish the peer review history of their article (what does this mean?). If published, this will include your full peer review and any attached files.

Reviewer #1: No

Reviewer #2: No

Reviewer #3: No

Reviewer #4: No

Reviewer #5: No

---

## [Author Response · Author response to Decision Letter 0]

7 Mar 2021

Manuscript PONE-D-21-03081

Response to Editor and Reviewers 

Thank you for the valued comments, and the effort made to improve our manuscript titled, Knowledge, attitude and practice of cervical cancer screening among women infected with HIV in Africa: Systematic review and meta-analysis. We are very grateful for the insightful comments and the input made. Based on the valuable suggestions forwarded, we revised the MS and incorporated all the comments as appropriate and highlighted the changes made. 

Regarding the suggestion on grammatical errors, senior instructors and co-authors critically reviewed, and online Spell checker – grammar and spell check in English - Reverso was used to check the spelling and grammar of the manuscript submitted. As commented by all reviewers, the entire manuscript is reassessed and minor corrections are made accordingly in the revised version. It is a privilege to work with you in the future prospects considering quality of the review and hard work. 

The followings are the line by line responses made for the valued suggestions and comments raised by your esteemed Editors and reviewers;

Editor comments: 

Comment 1. Check PLOS ONE's style requirements

• Authors' response: Thank you for pointing out this issue. The revised manuscript followed PLOS ONE manuscript formatting guidelines, including use of level 1 heading for all major sections, bold type, 18pt font. 

Comment 2. Confirm that you have included all items recommended in the PRISMA checklist

• Authors' response: Thank you. We confirmed that all the items in the PRISMA checklist are included and the search strategy is well described on page 5 of the manuscript. 

• We tried to restrict the studies done within the last 10 years since we were interested to assess the recent/current knowledge, attitude and practice of HIV positive women towards cervical cancer screening. 

• The results of the individual components of quality assessment are included as a supplemental file (S2 Table) as suggested. 

Comment 3. Include a table as part of main manuscript. 

• Authors' response: Thank you and we have inserted the tables as suggested in the revised version of the manuscript [page 8]

Comment 4. State the financial disclosure

• Authors' response. Comment incorporated [Page 16].

Comment 5. PLOS requires an ORCID iD for the corresponding author

• Author response. Thank you. Comment incorporated and the ORCID iD for the corresponding author's is 0000-0002-2794-8251

Comment 6. Your ethics statement should only appear in the Methods section of your manuscript

• Authors' response: Thank and the comments is incorporated. [Page 9]

Comment 7. Include captions for your Supporting Information files at the end of your manuscript

• Authors' response: Comment incorporated. 

Reviewers' Comments: 

Reviewers' Comments Authors response Remark

Reviewer #1 Grammatical errors Done as suggested. 

Definite time frame for search 

 The search is already filtered by selecting 10 years in the left corner of database search till the time of data extraction which is August 2020. 

 The number of articles identified do not add-up to 131 Thank you for pointing this out. The records identified via Google scholar (n=46) are covered due to font size and now visible. 

Reviewer #2 -I will worry about disparity in healthcare access creating a bias on KAP

- Most patients were from Ethiopia and the difference in health system from different countries can affect the generalizability of the result to whole of the African Continent

- Cervical cancer incidence correlate with a lower CD4 count Thank you very much for your valuable comments and suggestions. 

-Our review was conducted among HIV population from ART clinics with more or less similar characteristics in terms of awareness and practice. 

- Only two articles are included in the analysis from Ethiopia, which fulfilled the eligibility criteria. [Table 1][Page 8]

- Sure! Immunosuppression hasan association with cervical cancer, but our aim here was to pool KAP estimates on cervical cancer among HIV population. 

Reviewer #3 -Define in the methodology what they meant by knowledge, attitude and practice (KAP)

 Thank you for your valuable comments and suggestions. 

- We included the operational definition of KAP under the methods section [Page 9].

- Other comments are also included in the revised manuscript.

- 

Reviewer #4 What are the specific parameters you used to decide whether KAP is prevalent or not? Our aim in this regard is to pool the pocket studies which reported the proportions of KAP among HIV positive women. Based on this finding, our next step is to use structured questionnaire to address KAP issue in the subsequent original study and establish the average score. 

Other comments also incorporated into the protocol.

Reviewer #5 It will be informative to investigate the reason of such existing heterogeneity in knowledge, attitude and practice of cervical cancer screening among women infected with HIV in Africa.

 Thank you very much for your valuable comment. 

-We tried to use meta-regression to rule out heterogeneity on KAP considering variation in the sample size and years of publication might cause heterogeneity observed. [Page 11] and as you can see from Table 1, age, even reported in age range, mean age, etc., which is very difficult to run a meta-regression to rule-out heterogeneity due to age. 

Thank you very much, 

Authors'

---

## [Decision Letter · Decision Letter 1]

29 Mar 2021

Knowledge, attitude and practice of cervical cancer screening among women infected with HIV in Africa: Systematic review and meta-analysis

PONE-D-21-03081R1

Dear Dr. Bogale

We’re pleased to inform you that your manuscript has been judged scientifically suitable for publication and will be formally accepted for publication once it meets all outstanding technical requirements.

Kind regards,

Saeed Ahmed, MD

Academic Editor

PLOS ONE

---

## [Editor Report · Acceptance letter]

31 Mar 2021

PONE-D-21-03081R1 

Knowledge, attitude and practice of cervical cancer screening among women infected with HIV in Africa: Systematic review and meta-analysis 

Dear Dr. Bogale:

I'm pleased to inform you that your manuscript has been deemed suitable for publication in PLOS ONE. Congratulations! Your manuscript is now with our production department. 

Kind regards, 

on behalf of

Dr. Saeed Ahmed 

Academic Editor

PLOS ONE